# Robust Multimodal Large Language Models Against Modality Conflict

**Zongmeng Zhang** [1]   **Wengang Zhou** [2]   **Jie Zhao** [3]   **Houqiang Li** [2]

## Abstract

Despite the impressive capabilities of multimodal large language models (MLLMs) in vision-language tasks, they are prone to hallucinations in real-world scenarios. This paper investigates the hallucination phenomenon in MLLMs from the perspective of **modality conflict**. Unlike existing works focusing on the conflicts between model responses and inputs, we study the inherent conflicts in inputs from different modalities that place MLLMs in a dilemma and directly lead to hallucinations. We formally define the modality conflict and construct a dataset named Multimodal Modality Conflict (MMMC) to simulate this phenomenon in vision-language tasks. Three methods based on prompt engineering, supervised fine-tuning, and reinforcement learning are proposed to alleviate the hallucination caused by modality conflict. Extensive experiments are conducted on the MMMC dataset to analyze the merits and demerits of these methods. Our results show that the reinforcement learning method achieves the best performance in mitigating the hallucination under modality conflict, while the supervised fine-tuning method shows promising and stable performance. Our work sheds light on the unnoticed modality conflict that leads to hallucinations and provides more insights into the robustness of MLLMs. The code and dataset are available at https://github.com/zmzhang2000/MMMC.

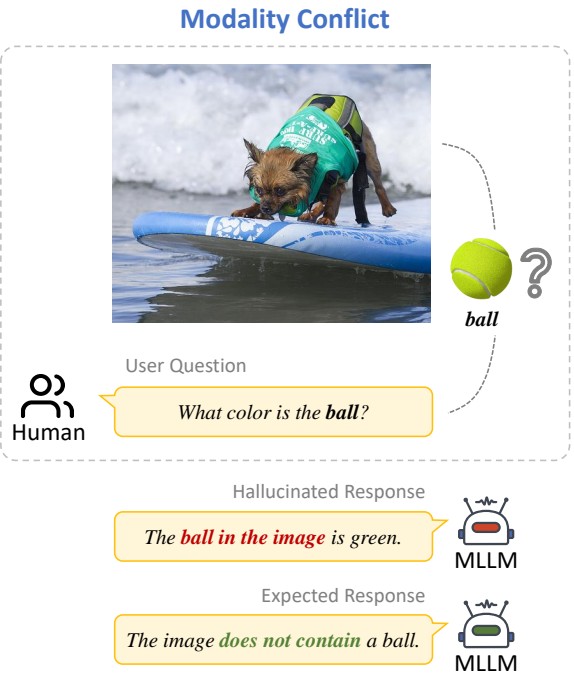

*Figure 1.* An example of modality conflict in vision-language tasks. Given an image describing a dog surfing on the sea, the user may ask the question "*What color is the ball?*". The model may hallucinate a response "*The ball in the image is green*", while there is no ball in the image. We expect the model to recognize the conflict between the visual input and the textual input and give a response like "*The image does not contain a ball*".

## 1. Introduction

The recent success of multimodal large language models (MLLMs) has advanced the development of artificial intelligence in vision-language tasks (Dai et al., 2023; Liu et al., 2023; 2024b; Bai et al., 2023; Wang et al., 2024). These models enable the joint reasoning over visual and textual inputs, and have achieved state-of-the-art performance in various vision-language tasks that require multimodal reasoning (Fu et al., 2024; Yue et al., 2024; Yu et al., 2024c; Lu et al., 2024; Liu et al., 2025). The powerful capabilities of these MLLMs are typically achieved by pretraining separate language and vision models on large-scale datasets, and then

---

[1]School of Artificial Intelligence and Data Science, University of Science and Technology of China [2]Department of Electronic Engineering and Information Science, University of Science and Technology of China [3]Huawei Technologies Co., Ltd.. Correspondence to: Wengang Zhou <zhwg@ustc.edu.cn>.

*Proceedings of the 42nd International Conference on Machine Learning*, Vancouver, Canada. PMLR 267, 2025. Copyright 2025 by the author(s).

aligning their features to enable multimodal reasoning (Liu et al., 2023; 2024b).

Despite the impressive performance of MLLMs, they are prone to hallucinations in real-world scenarios (Huang et al., 2024; Yu et al., 2024b). Hallucinations refer to the phenomenon where MLLMs generate incorrect or misleading information not supported by the input data (Ji et al., 2023). Existing works have proposed various methods to alleviate hallucinations in MLLMs, such as improving the quality of training data (Liu et al., 2024a; Yu et al., 2024a), adjusting the decoding strategies (Leng et al., 2024; Huang et al., 2024), and align the model with human preference (Zhao et al., 2024; Yu et al., 2024b). These methods mainly target more precise alignment between the features of different modalities to reduce hallucinations.

However, existing works on alleviating hallucinations in MLLMs mainly focus on the conflicts between the model responses and the inputs, neglecting a possible source of hallucinations: the conflicts between the inputs from different modalities, which we call **modality conflict**. For instance, as shown in Figure 1, given an image describing a dog surfing on the sea, the user may ask the question "*What color is the ball?*". In this case, the question supposes a ball exists in the image, and the model may hallucinate a response "*The ball in the image is green*", while there is no ball in the image. We expect the model to recognize the conflict between the visual input and the textual input and give a response like "*The image does not contain a ball*". Even with the capability of perfectly aligning features of different modalities, MLLMs may still fall into a dilemma when facing such intrinsically conflicted information between inputs. To this end, we aim to investigate such hallucination phenomenon in MLLMs from the perspective of modality conflict.

In this paper, we first give a formal definition of modality conflict in vision-language tasks in terms of objects, attributes, and relationships in the visual and textual inputs. Based on the definition, we construct a dataset named Multi-Modal Modality Conflict (MMMC) to simulate the modality conflict in vision-language tasks. We evaluate various prevalent MLLMs (Dai et al., 2023; Liu et al., 2024b; Bai et al., 2023; Wang et al., 2024) on the MMMC dataset and find that most of them lack the ability to recognize the modality conflict and are prone to hallucinations.

To alleviate the hallucination caused by the modality conflict and work towards more robust MLLMs, we investigate the effectiveness of three methods: prompt engineering, supervised fine-tuning, and reinforcement learning. We conduct extensive experiments on the MMMC dataset to analyze the merits and demerits of these methods. Our results show that the reinforcement learning method achieves the best performance in mitigating the hallucination under modality conflict, while the supervised fine-tuning method shows

promising and stable performance. Our work sheds light on the unnoticed modality conflict that causes hallucinations and provides more insights into the robustness of MLLMs.

To summarize, the contributions of this paper are as follows:

- This paper reveals an unnoticed source of hallucinations in MLLMs: modality conflict. The formal definition of modality conflict is presented in the level of objects, attributes, and relationships.

- We construct a dataset called Multimodal Modality Conflict (MMMC) to simulate the modality conflict in vision-language tasks and evaluate various prevalent MLLMs on the dataset. Results show that most MLLMs are prone to hallucinations under modality conflict.

- We propose three methods, prompt engineering, supervised fine-tuning, and reinforcement learning, to alleviate the hallucination caused by the modality conflict. Extensive experiments are conducted to analyze the merits and demerits of these methods.

## 2. Problem Formulation

In this section, we formally define modality conflict in vision-language tasks and detail the data construction process of MMMC. The pipeline of the data construction process and proposed methods are illustrated in Figure 2.

### 2.1. Modality Conflict

**General Form**  Given a vision-language task consisting of a visual input $\mathcal{V}$ and a textual input $\mathcal{T}$, the task is to predict an answer $\mathcal{A}$. We define the modality conflict as the situation where the information contained in $\mathcal{V}$ and $\mathcal{T}$ is inconsistent with each other, leading to a dilemma for the model to predict the answer $\mathcal{A}$. We define the general form of modality conflict as

$$\text{Info}(\mathcal{V}) \neq \text{Info}(\mathcal{T}). \tag{1}$$

Concretely, we instantiate the $\text{Info}(\cdot)$ function from objects, attributes, and relationships in the visual and textual inputs following Shu et al. (2025). We define these three types of modality conflict as follows.

**Object Conflict**  The object conflict occurs when the textual input involves objects not present in the visual input. For example, the textual input supposes *a cat* in the image, while the image only contains *a dog* rather than *a cat*. We define the object conflict in $\langle \mathcal{V}, \mathcal{T} \rangle$ as

$$\text{Obj}(\mathcal{T}) \nsubseteq \text{Obj}(\mathcal{V}), \tag{2}$$

where $\text{Obj}(\cdot)$ denotes the set of objects in the input.

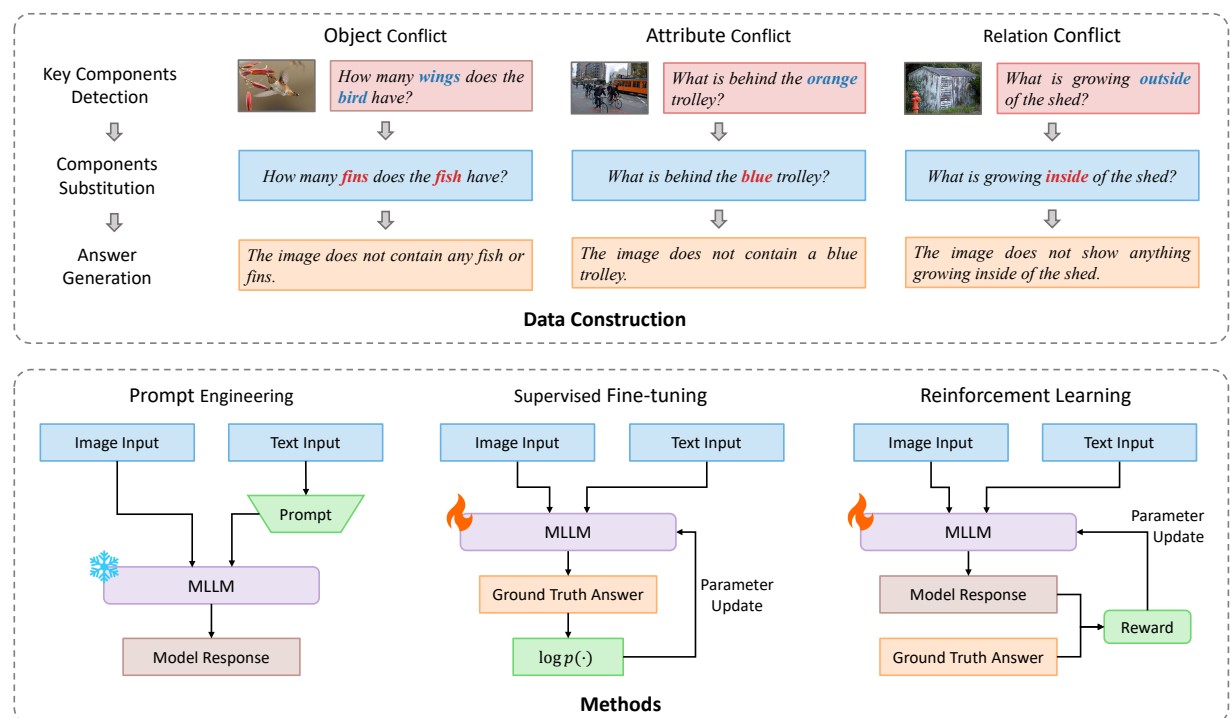

*Figure 2.* The pipeline of the data construction process and proposed methods. The data construction process mainly consists of key components detection, components substitution, and answer generation. Prompt engineering, supervised fine-tuning, and reinforcement learning are proposed to alleviate the hallucination caused by the modality conflict. The snowflake icon denotes that the MLLM is frozen, while the flame icon indicates it is fine-tuned.

**Attribute Conflict**   Sometimes the visual and textual inputs may describe the same objects but with different attributes. For example, the textual input describes *a red apple*, while the image shows *a green apple*. We deem attribute conflict arises in $\langle \mathcal{V}, \mathcal{T} \rangle$ if

$$\begin{cases} \mathrm{Obj}(\mathcal{T}) \subseteq \mathrm{Obj}(\mathcal{V}) \\ \{\mathcal{O}_i\}_{i=1}^m = \mathrm{Obj}(\mathcal{T}) \cap \mathrm{Obj}(\mathcal{V}) \\ \mathrm{Attr}(\mathcal{O}_i^{\mathcal{T}}) \neq \mathrm{Attr}(\mathcal{O}_i^{\mathcal{V}}), i = 1, 2, ..., m \end{cases}, \quad (3)$$

where $\{\mathcal{O}_i\}_{i=1}^m$ is the set of objects contained in both the image and text inputs. $\mathcal{O}_i^{\mathcal{V}}$ and $\mathcal{O}_i^{\mathcal{T}}$ indicate the objects in image and text inputs, respectively. $\mathrm{Attr}(\cdot)$ denotes the attributes of an object.

**Relationship Conflict**   The relationship conflict occurs when the visual and textual inputs describe the same objects with different relationships. For example, the textual input describes *a cat on the table*, while the image shows *a cat on the floor*. We formulate the relationship conflict in $\langle \mathcal{V}, \mathcal{T} \rangle$

as a situation where

$$\begin{cases} \mathrm{Obj}(\mathcal{T}) \subseteq \mathrm{Obj}(\mathcal{V}) \\ \{\mathcal{O}_i\}_{i=1}^m = \mathrm{Obj}(\mathcal{T}) \cap \mathrm{Obj}(\mathcal{V}) \\ \mathrm{Rel}(\mathcal{O}_i^{\mathcal{T}}, \mathcal{O}_j^{\mathcal{T}}) \neq \mathrm{Rel}(\mathcal{O}_i^{\mathcal{V}}, \mathcal{O}_j^{\mathcal{V}}), i, j = 1, 2, ..., m \end{cases}, \quad (4)$$

where $\mathrm{Rel}(\cdot)$ denotes the relationships between two objects.

### 2.2. Data Construction

To simulate the modality conflict in vision-language tasks, we construct a dataset Multimodal Modality Conflict (MMMC) that contains all the three types of conflicts discussed above. Specifically, we collect images from the widely-used vision-language datasets, Visual Genome (Krishna et al., 2017), and construct natural language questions conflicting with the image content and corresponding answers. Given the clear definition of modality conflict, we resort to the large language models[1] to construct the dataset for modality conflict. The construction process is elaborated as follows.

---

[1]We use GPT-4o-mini, a powerful and fast model for data construction.

**Base Question Sampling** To align the format and style of questions to the original dataset, we adopt a substitution framework to simulate the modality conflict, inspired by Longpre et al. (2021). We first randomly sample a base question $\mathcal{T}$ from the original dataset for each image $\mathcal{V}$. Key information in the base question will then be substituted with conflicting ones to construct a new question as discussed in the following.

**Key Components Detection** Questions in vision-language tasks usually involve a series of components, including objects, attributes, and relationships. These components should be displayed in the image to ensure the question is answerable. However, in the modality conflict scenario, the components in the question may not be present in the image. We adopt the large language model to detect the objects in the image and extract the attributes and relationships of the objects.

**Components Substitution** We substitute the objects, attributes, and relationships in the base question with conflicting information detected from the image. The substitution process is conducted by directly prompting a large language model to generate a counterfactual question according to the original question and the key components to be substituted. Additionally, we input all the objects, attributes, and relationships in the image, from the annotation of the original dataset to the model to ensure the conflict between the question and the image content.

**Answer Generation** After obtaining the conflicting question $\mathcal{T}'$, we generate a paired answer $\mathcal{A}'$ for the question. Unlike existing works that generate multiple-choice answers (Zhu et al., 2024), we collect model response directly to improve the model robustness in free-form generation. It is worth noting that, to avoid the impact of hallucinations in the widely-used large vision-language models, we do not directly generate the answer by inputting the image $\mathcal{V}$ and the question $\mathcal{T}'$ to them. Instead, we impose that the substituted components in the question are not present in the image on the large language model, and require the model to generate the answer $\mathcal{A}'$ based on the conflicting information. The large language model demonstrates the capability of generating answers based barely on textual information, as shown in Figure 2.

**Postprocessing** These generated questions and answers are then verified by human annotators to ensure the quality of the dataset. The language fluency, the conflict between the question and the image, and the correctness of the answer are all considered in the verification process. Finally, we obtain 20K image-question-answer triples in the MMMC dataset and randomly split them into 18K training samples and 2K testing samples. Visualizations for the statistics of

MMMC is provided in Appendix A.

## 3. Method

We propose three methods, *i.e.* prompt engineering, supervised fine-tuning, and reinforcement learning, to alleviate the hallucination caused by the modality conflict. We first formulate the vision-language task as a conditional generation problem:

$$\mathcal{A} \sim \pi_\theta(\mathcal{A}|\mathcal{V}, \mathcal{T}) = \sum_{t=1}^{T} \pi_\theta(a_t|\mathcal{V}, \mathcal{T}, a_{<t}), \qquad (5)$$

where the model $\pi_\theta$ is required to sequentially generate the answer $\mathcal{A}$ given the visual input $\mathcal{V}$ and the textual input $\mathcal{T}$, and $T$ is the length of answer $\mathcal{A}$. We then introduce the three methods to improve the robustness of the model against modality conflict in this section.

### 3.1. Prompt Engineering

Instruction following (Dai et al., 2023; Liu et al., 2023; 2024b) is a fundamental capability of MLLMs. Questions are directly inputted to the model to guide the generation of the answer in Equation (5). We propose to instruct the model to check if the objects, attributes, and relationships in the question are present in the image before generating the answer with a simple but effective prompt template $p(\mathcal{T})$:

> *Please check if the image contains mentioned information and answer the question:* $\mathcal{T}$

The prompt engineering method is easy to implement and does not require additional data or computational resources, formulated as

$$\mathcal{A} \sim \pi_\theta(\mathcal{A}|\mathcal{V}, p(\mathcal{T})). \qquad (6)$$

However, the prompt engineering method may not be effective in all cases. The performance of the method is heavily dependent on the foundation model and the quality of the prompt. Besides, the potential of training data is not exploited in the prompt engineering method. Therefore, we explore methods with additional training to fully leverage the data and improve the robustness of the model against modality conflict.

### 3.2. Supervised Fine-Tuning

Supervised fine-tuning aims to learn a mapping from the input to the output by minimizing the discrepancy between the model predictions and the ground-truth labels. Existing works have shown the superiority of supervised fine-tuning in conquering knowledge conflict in LLMs (Longpre et al., 2021).

We propose to fine-tune the model on the MMMC dataset with the language modeling objective, formulated as

$$\pi_\theta^* = \arg\min_\theta \mathbb{E}_{\langle \mathcal{V}, \mathcal{T}, \mathcal{A} \rangle \sim \mathcal{D}} \left[ -\log \pi_\theta(\mathcal{A}|\mathcal{V}, \mathcal{T}) \right], \quad (7)$$

where $\langle \mathcal{V}, \mathcal{T}, \mathcal{A} \rangle$ is a triplet of image, question, and answer in the MMMC dataset $\mathcal{D}$. With this objective, the model is optimized by gradient descent to align the model predictions with the ground-truth labels, which is expected to improve the robustness of the model against modality conflict.

Despite its effectiveness, supervised fine-tuning mainly emphasizes adapting the style of the model to the target domain (Zhou et al., 2023), while the performance improvement on the unseen data may be limited.

### 3.3. Reinforcement Learning

Inspired by the success of reinforcement learning in alignment with human preference (Ouyang et al., 2022; Stiennon et al., 2020; Yu et al., 2024b) and improving the robustness of large language models (Zhang et al., 2024), we resort to reinforcement learning to further improve the robustness of the model against modality conflict. Specifically, the conditional generation problem in Equation (5) can be formulated as a Markov Decision Process (MDP):

$$\mathcal{A} \sim \pi_\theta(\mathcal{A}|\mathcal{V}, \mathcal{T}) \Leftrightarrow \langle S, A, r, P, \rho_0, \gamma \rangle, \quad (8)$$

with the state $s_t = (\mathcal{V}, \mathcal{T}, a_{<t})$, the action $a_t$, the reward $r_t$, the transition probability $P(s_{t+1}|s_t, a_t)$, the initial state distribution $\rho_0(s_0) : \langle \mathcal{V}, \mathcal{T} \rangle \sim \mathcal{D}$, and the discount factor $\gamma$. We propose to optimize the model with reinforcement learning by maximizing the expected reward, formulated as

$$\pi_\theta^* = \arg\max_\theta \mathbb{E}_{s_0 \sim \rho_0} \mathbb{E}_{a_t \sim \pi_\theta(a_t|s_t)} \left[ \sum_{t=1}^{T} \gamma^t r_t \right]. \quad (9)$$

To realize the goal of alleviating the hallucination caused by the modality conflict, we assign a reward function that encourages the model to generate answers semantically consistent with the one in the MMMC dataset and penalizes the model for generating hallucinated responses. The reward function is defined as

$$r_t = \begin{cases} +1, & \text{if } t = T \wedge a_{\leq t} \text{ is consistent with } \mathcal{A} \\ -1, & \text{if } t = T \wedge a_{\leq t} \text{ is not consistent with } \mathcal{A} , \\ 0, & \text{otherwise} \end{cases}$$

(10)

where $a_{\leq t}$ denotes the generated answer at time step $t$ and $\mathcal{A}$ is the ground-truth answer in the MMMC dataset. We prompt a pretrained large language model to judge the semantic consistency between the generated and ground-truth answer and assign the reward based on the judgment. Detailed prompts are listed in Appendix B.

With these base components, we can optimize the model with arbitrary reinforcement learning algorithms, such as Proximal Policy Optimization (PPO) (Schulman et al., 2017) and REINFORCE (Williams, 1992). We adopt an optimized version of REINFORCE algorithm, REINFORCE++ (Hu, 2025), for light computation and good performance.

In the reinforcement learning method, the model is optimized by interacting with the environment and receiving rewards based on the quality of the generated answers. Due to the nature of sampling data from the model itself, the reinforcement learning method is expected to learn more diverse and robust answers that share similar semantics with the ground-truth answers.

## 4. Experiments

### 4.1. Setup

**Models** We evaluate several types of prevalent MLLMs, InstructBLIP (Dai et al., 2023), LlaVA-v1.5 (Liu et al., 2023), LLaVA-NeXT (Liu et al., 2024b), and Qwen2-VL-Instruct (Wang et al., 2024) series, on the MMMC dataset. InstructBLIP adopts Q-former (Li et al., 2023) to compress the image into 32 tokens and bridge the vision and language features, while LLaVA-v1.5, LLaVA-NeXT and Qwen2-VL-Instruct separately encode the image and text with transformer architecture and conduct multimodal reasoning with additional adapter modules. We use 7B version for each model in the evaluation. Besides, to investigate the impact of model size, we also evaluate the 2B version of Qwen2-VL-Instruct. Additionally, we include the widely-used large language model GPT-4o as a baseline.

**Implementation Details** We implement all proposed method using Hugging Face Transformers and OpenRLHF library (Hu et al., 2024). For the supervised fine-tuning, we use the Adam optimizer with a learning rate of $5 \times 10^{-6}$ and a batch size of 8. We train the model for 1 epochs on the MMMC dataset with 10000 training samples except for the ablation study. For the reinforcement learning, we use the Adam optimizer with a learning rate of $9.65 \times 10^{-6}$ and a batch size of 8. We train the model on the MMMC dataset with only 1000 training samples since longer reinforcement learning will cause the model collapse. We set the KL coefficient to 0.01 and the max response length to 128. Both the supervised fine-tuning and reinforcement learning methods are trained with LoRA (Hu et al., 2021). We use the Llama-3.3-70B-Instruct for reward model.

**Evaluation Protocol** Given the reference responses in the MMMC dataset, we adopt the widely-used ROUGE-L (Lin, 2004) F-measure to evaluate the longest common subsequences overlap between the model responses and the reference responses. However, this traditional metric do not

*Table 1.* Explanation for each level of overall quality scores in the LLM-as-a-Judge evaluation.

| Score | Quality | Detailed Description |
|---|---|---|
| 0 | Not Valid | Unnatural, incoherent or unreadable |
| 1 | Terrible | Irrelevant to the question asked |
| 2 | Wrong | Different from the reference answer, but still relevant to the question |
| 3 | Right | Has the same meaning as the reference, but may be phrased differently |
| 4 | Excellent | Same as the reference or more naturally |

consider the semantic similarity in language precisely.

To present a more intuitive evaluation, we adopt the LLM-as-a-Judge (Zheng et al., 2023), a large language model that is pretrained on a large-scale dataset and fine-tuned on human preference data, to evaluate the quality of the model responses. Concretely, to evaluate the robustness of the model against modality conflict, we calculate the hallucination rate (Hallu-Rate), defined as the percentage of hallucinated responses in the model responses. A response is considered hallucinated if it erroneously assumes the existence of objects, attributes, or relationships that not present in the image, presenting plausible but incorrect information.

Additionally, we require LLM-judge to evaluate the overall quality of the model responses concerning fluency, relevance, and correctness, represented by a score ranging from 0 to 4. We list the criteria for these scores in Table 1. The average scores of the LLM-judge are reported as LLM-Judge. To obtain more robust results, we adopt strong closed-source model GPT-4o series and open-source model Llama-3.3-70B to perform evaluations of Hallu-Rate and LLM-Judge. All prompts we used in the evaluation are listed in Appendix B.

### 4.2. Main Results

**Robustness of Prevalent Foundation Models Against modality conflict** As the "Base" results in Section 4.2 show, the prevalent MLLMs, InstructBLIP-7B, LLaVA-v1.5-7B, LLaVA-NeXT-7B, Qwen2-VL-Instruct-7B and even GPT-4o, perform poorly on the MMMC dataset. All models exhibit Hallu-Rate over 40%, indicating that they are prone to hallucinations under modality conflict. The LLM-Judge scores are lower than 2.5, showing that most of their responses are judged as wrong or of lower quality. The ROUGE-L scores are also relatively low, suggesting that the model responses are not sufficiently aligned with the reference responses.

**Performance Improvements with Proposed Methods** We then evaluate the effectiveness of the proposed meth-ods, prompt engineering, supervised fine-tuning, and reinforcement learning, on the MMMC dataset. As shown in Section 4.2, these three methods significantly improve the robustness of the prevalent MLLMs on the MMMC dataset. The Hallu-Rate is reduced by 10% to 50% with the proposed methods. The LLM-Judge scores are improved by 0.4 to 0.9, indicating that the overall quality of the model responses is enhanced. The consistent conclusions provided by the LLM judge, which is based on GPT-4o and Llama-3.3-70B, further validate the reliability of our evaluation results. The ROUGE-L scores are also improved with several methods, showing that the model responses are more aligned with the reference responses. We will further analyze the merits and demerits of these methods in the following section.

### 4.3. Analysis

**Further Analysis on Proposed Methods** Prompt engineering is a basic method that may improve the robustness of the model responses against modality conflict, bring reduced Hallu-Rate, and improve LLM-Judge scores in most cases. It is easy to implement and does not require additional data or computational resources. However, as shown in Section 4.2, the improvement of prompt engineering is heavily dependent on the foundation model, and the performance may be unstable. Prompt engineering brings significant improvement to Qwen2-VL-Instruct-7B and LLaVA series, but increases the Hallu-Rate of smaller Qwen2-VL-Instruct-2B model. The performance on InstructBLIP is nearly the same as the base model. We inspect the generated responses of InstructBLIP and find that the model tends to generate short and simple responses whatever the prompt is, which may lead to the limited improvement of prompt engineering. The effectiveness of prompt engineering is also limited on GPT-4o due to the over-robustness of the model, with which the model tends to generate similar responses for different expressions of the same instruction.

Supervised fine-tuning (SFT) is a more advanced method that can further improve the robustness of the model responses. It requires additional data and computational resources for fine-tuning, but it can achieve better performance than prompt engineering. As shown in Section 4.2, SFT reduces the Hallu-Rate and improves the LLM-Judge scores of LLaVA-v1.5, LLaVA-NeXT and Qwen2-VL-Instruct series. However, InstructBLIP suffers from performance loss after SFT. We speculate that the pre-training of InstructBLIP does not inject the capability of recognizing the conflicts between modalities and the fine-tuning data in MMMC is also not enough to teach this new skill.

Besides, SFT restricts the model behavior to the fine-tuning data, which may lead to overfitting and limited generalization. Reinforcement learning (RL) samples the responses from the model itself and provides more diverse and infor-

*Table 2.* Performance comparison of different methods on the MMMC dataset. We conduct the experiments on Prompt Engineering (PE), Supervised Fine-Tuning (SFT), and Reinforcement Learning (RL). The performance of GPT-4o is also reported for comparison. The results of SFT and RL are averaged across three runs with different seeds, and the standard deviations are reported in parentheses. ↑ denotes the higher the better, while ↓ denotes the lower the better. The best performance for each model is highlighted in bold.

| Model | Method | ROUGE-L (%) ↑ | Hallu-Rate (%) ↓ (Llama) | Hallu-Rate (%) ↓ (GPT) | LLM-Judge ↑ (Llama) | LLM-Judge ↑ (GPT) |
|---|---|---|---|---|---|---|
| GPT-4o | Base | 23.76 | **59.40** | 57.00 | 2.12 | 2.39 |
| | PE | **23.98** | 60.10 | **56.95** | **2.13** | **2.42** |
| InstructBLIP-7B | Base | **13.89** | 82.10 | 70.55 | 1.81 | 1.85 |
| | PE | **13.89** | 82.30 | 69.50 | 1.79 | **1.86** |
| | SFT | 8.86 (0.43) | 85.48 (0.37) | 70.33 (0.74) | **1.81** (0.01) | 1.76 (0.05) |
| | RL | 5.65 (3.08) | **57.62** (18.58) | **57.18** (18.07) | 1.01 (0.69) | 1.48 (0.96) |
| LLaVA-v1.5-7B | Base | **28.54** | 93.25 | 83.60 | 1.73 | 1.81 |
| | PE | 25.83 | 86.95 | 84.70 | 1.94 | 1.93 |
| | SFT | 16.90 (0.52) | 59.37 (1.02) | 52.28 (0.92) | 2.27 (0.02) | 2.27 (0.02) |
| | RL | 23.53 (3.49) | **33.87** (2.53) | **29.78** (2.04) | **2.58** (0.04) | **2.74** (0.04) |
| LLaVA-NeXT-7B | Base | 18.08 | 69.65 | 67.00 | 1.92 | 2.24 |
| | PE | 20.91 | 50.50 | 50.00 | 2.43 | 2.69 |
| | SFT | 22.25 (0.07) | 45.93 (0.47) | 42.83 (0.78) | 2.48 (0.01) | 2.44 (0.04) |
| | RL | **25.52** (1.66) | **33.83** (1.99) | **31.27** (2.03) | **2.65** (0.02) | **2.86** (0.05) |
| Qwen2-VL-Instruct-2B | Base | 25.20 | 46.55 | 40.55 | 2.07 | 2.26 |
| | PE | **30.12** | 62.10 | 59.95 | 2.26 | 2.40 |
| | SFT | 29.32 (0.25) | 26.85 (0.60) | 32.78 (0.74) | 2.71 (0.02) | 2.76 (0.02) |
| | RL | 22.65 (1.65) | **18.00** (5.19) | **16.78** (4.30) | **2.73** (0.06) | **2.97** (0.08) |
| Qwen2-VL-Instruct-7B | Base | 24.73 | 52.35 | 47.95 | 2.25 | 2.47 |
| | PE | **28.65** | 40.10 | 37.35 | 2.52 | 2.80 |
| | SFT | 28.60 (0.10) | 28.58 (0.34) | 32.02 (0.69) | **2.71** (0.01) | 2.74 (0.02) |
| | RL | 18.89 (0.82) | **23.52** (5.63) | **20.45** (5.09) | 2.66 (0.07) | **2.86** (0.10) |

mative data for training. It requires more computational resources but may achieve better performance than SFT. As shown in Section 4.2, RL dramatically reduces the Hallu-Rate and improves the LLM-Judge scores of all models, especially on Qwen2-VL-Instruct series. The main reason for the best performance of RL is that it explores more diverse responses in the training process than SFT, which helps the model to recognize the conflicts between modalities and enhance its robustness.

**Performance Breakdowns for Different Conflict Types** In order to gain a deeper understanding of how each approach tackles various conflict types, we provide a detailed performance analysis for each category of conflict in Appendix C. The analyses indicate that the conclusions drawn from individual subsets of conflict types align closely with those derived from the entire dataset. Particularly noteworthy is the finding that MLLMs exhibit superior performance on object-conflict types. On the other hand, attribute-conflict scenarios present a moderate level of difficulty, and relationship-conflict types pose a significant challenge for MLLMs. The performance on these conflicts is notably poorer when compared to object and attribute conflicts. This drop in performance can be attributed to the intricate rela-

tional dynamics that the models struggle to accurately interpret and predict. These observations suggest that while substantial advancements have been made in processing simpler conflict types such as object-conflicts, there remains a critical need for enhancement in managing relationship-focused conflicts.

**Alignment Tax** Both the SFT and RL methods require parameter updates to align the model with the training data, and thus introduce the alignment tax, which is defined as the performance loss of the model on the original task after the fine-tuning (Ouyang et al., 2022). To analyze the alignment tax of our methods, we test the performance changes of the models on a wide range of vision-language tasks after the fine-tuning, including HallusionBench (Guan et al., 2024), MMBench (Liu et al., 2025), MMStar (Chen et al., 2024), MMMU (Yue et al., 2024), MathVista (Lu et al., 2024), OCRBench (Liu et al., 2024d), AI2D (Kembhavi et al., 2016), MMVet (Yu et al., 2024c) and MME (Fu et al., 2024).

We visualize the performance change of the SFT and RL methods in Figure 3. As the figure shows, the alignment tax is also heavily dependent on the foundation model. For example, InstructBLIP suffers a lot from the alignment tax, showing half of the performance loss on MMBench with

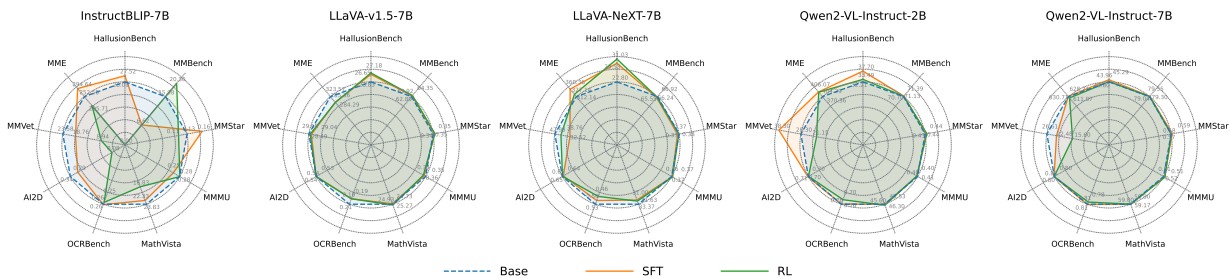

*Figure 3.* Visualization of the alignment tax for supervised fine-tuning (SFT) and reinforcement learning (RL). We plot the performance of the base model (Base) with blue dashed regular polygon, and the performance of the SFT and RL models with orange and green solid polygons, respectively. All scores are normalized to the base model for intuitive comparison.

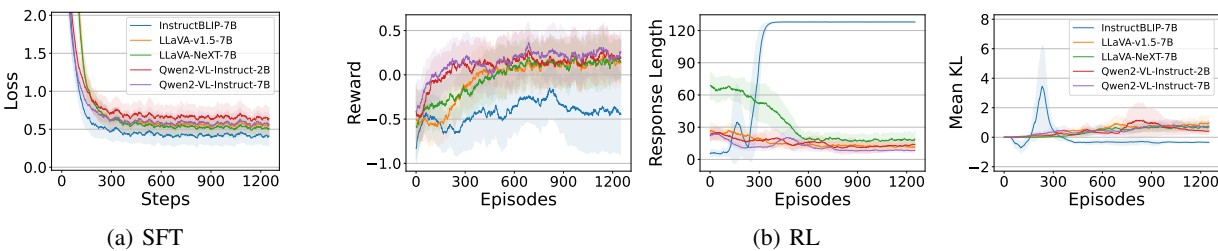

|  (a) SFT | (b) RL |
|---|---|

*Figure 4.* Training curves of supervised fine-tuning (SFT) and reinforcement learning (RL) on the MMMC dataset. The training loss of SFT, reward, response length and mean KL divergence of RL are plotted. We plot the average training curves over three runs with different seeds. The solid lines and shaded areas represent the mean values and standard deviations, respectively. All curves are smoothed with exponential moving average for better visualization.

SFT and three-quarters of the performance loss on AI2D and MMVet with RL. By contrast, Qwen2-VL-Instruct-7B series shows negligible performance change on most tasks after the fine-tuning, indicating that the model is more robust to the alignment tax. Surprisingly, the performance of LLaVA-NeXT on HallusionBench is even improved after SFT and RL, and Qwen2-VL-Instruct-2B shows similar changes on MMVet with SFT. LLaVA-v1.5-7B is the most stable model, demonstrating consistent performance across multiple benchmarks without much variation. These results are also strong evidences that the capability of recognizing the conflicts between modalities is beneficial for the model to reduce other types of hallucinations.

**Training Stability** We further analyze the training stability of the SFT and RL methods on the MMMC dataset. As shown in Figure 4, in general, the training loss of SFT is relatively stable, while the reward of RL fluctuates a lot. The response length of RL is also unstable, indicating that the model may generate responses with different lengths during the training process. This clearly shows that although RL can potentially reach higher performance, it is less stable than SFT.

The most noticeable phenomenon is that InstructBLIP-7B's response length experiences a jump at around 300 episodes to the maximum length we set. Meanwhile, it is also the

time when the reward of RL reaches the valley bottom and the mean Kullback-Leibler (KL) divergence reaches a peak. To further investigate the reason for this phenomenon, we analyze the generated responses of InstructBLIP-7B and find that, at this time, the model begins to generate longer responses with tedious, repeated, and irrelevant information. We deem that the model may fall into the local optimum but is completely collapsed at that time. Some responses generated by the model are shown in Appendix D for a better understanding of the phenomenon. RL training curves of other models are more stable, showing that the model is well-trained with the reward signal. The fluctuation of the reward and response length may be caused by the exploration of the model in the training process, which helps the model to recognize the conflicts between different modalities and enhance its robustness.

**Impact of Training Episodes Number on RL** As the training of RL is unstable and prone to model collapse, we delve into a fundamental aspect of RL training: the impact of training episode number on the model performance. We conduct experiments on the LLaVA-NeXT-7B model with different training episodes and report the Hallu-Rate and LLM-Judge scores in Table 3. As the table shows, the Hallu-Rate is reduced by 20% with the increase of training episodes from 2000 to 10000, and the LLM-Judge scores are improved by 0.5. However, the performance is not

*Table 3.* Hallu-Rate and LLM-Judge of the LLaVA-NeXT-7B model with different training episodes on the MMMC dataset. Both Hullu-Rate and LLM-Judge are evaluate by GPT-4o series.

| Training Episodes | Hallu-Rate ↓ | LLM-Judge ↑ |
|---|---|---|
| 2000 | 63.90 | 2.31 |
| 5000 | 40.30 | 2.71 |
| 10000 | 28.50 | **2.86** |
| 20000 | 30.95 | 2.78 |
| 100000 | **25.55** | 2.81 |

further improved with more training episodes, indicating that the model may fall into the local optimum after 10000 episodes. We speculate that the model may need more diverse and informative data for training to further improve the robustness against modality conflict.

## 5. Related Work

### 5.1. Multimodal Large Language Models

With the significant progress of large language models, multimodal large language models (MLLMs) have been developed based on the language capabilities of large language models and the visual understanding of large vision models. Given the pretrained language and vision models, training of most MLLMs involves a pretraining stage to align the features of different modalities (Bai et al., 2023; Li et al., 2023; Alayrac et al., 2022), and a fine-tuning stage to inject the instruction following abilities into the model (Dai et al., 2023; Liu et al., 2024b; Guan et al., 2024; Liu et al., 2023). Despite their success in various vision-language tasks, MLLMs are prone to hallucinations (Ji et al., 2023), where the model generates content contradicting the input.

### 5.2. Hallucinations in MLLMs

Plenty of works have been proposed to alleviate hallucinations in MLLMs from the perspective of training data (Liu et al., 2024a; Yu et al., 2024a), decoding strategies (Leng et al., 2024; Huang et al., 2024), and human preference alignment (Zhao et al., 2024; Yu et al., 2024b). However, these efforts mainly focus on the conflicts between the model responses and the inputs, neglecting a possible source of hallucinations: the conflicts between the inputs from different modalities. Even with the capability of perfectly aligning features, MLLMs will fall into a dilemma when facing intrinsically conflicted information. This paper aims to investigate the hallucination phenomenon in MLLMs from the perspective of modality conflict.

### 5.3. Knowledge Conflict

Knowledge conflict (Xu et al., 2024; Longpre et al., 2021; Chen et al., 2022; Xie et al., 2024) is a long-discussed topic in the area of large language models. Longpre et al. (2021) formalizes the problem of knowledge conflicts between the contextual and the learned information. Chen et al. (2022) extends the problem to multiple source context scenarios and proposes a calibration model to detect the phenomenon. Analogically, conflicts emerge when inconsistent information is presented in multimodal tasks, leading to hallucinations in most MLLMs. Zhu et al. (2024) defines the problem of cross-modality parametric knowledge conflict, detects the problem with multiple-choice question answering, and proposes a dynamic contrastive decoding method to mitigate the impact of the conflicts. Liu et al. (2024c) specifies the contradiction between visual information and commonsense knowledge in the language model. These efforts in MLLMs neglect the impact of intrinsic conflict between modalities that lead MLLMs to hallucination. By contrast, we formalize the concept of modality conflict and collect a dataset to simulate this situation and evaluate the robustness of prevalent MLLMs against it.

## 6. Conclusion

In this paper, we investigate the hallucination phenomenon in multimodal large language models (MLLMs) from the perspective of modality conflict. We first give a formal definition of knowledge conflicts in vision-language tasks and construct a dataset, named MMMC. We then propose three methods, *i.e.* prompt engineering, supervised fine-tuning, and reinforcement learning, to alleviate the hallucination caused by the modality conflict. We evaluate the proposed methods on the MMMC dataset and analyze the results concerning the overlap with the reference responses, the hallucination rate, and the overall response quality. The results show that the proposed methods significantly improve the robustness of the prevalent MLLMs on the MMMC dataset. We further analyze the merits and demerits of these methods and provide more insights for future research.

## Acknowledgements

This work was supported by National Key R&D Program of China under Contract 2022ZD0119802, National Natural Science Foundation of China under Contract 623B2097, and the Youth Innovation Promotion Association CAS. It was also supported by the GPU cluster built by MCC Lab of Information Science and Technology Institution, USTC.

## Impact Statement

This paper presents work with a goal to advance the field of Machine Learning. There are many potential societal consequences of our work, none of which we feel must be specifically highlighted here.

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

# A. Visualization of MMMC

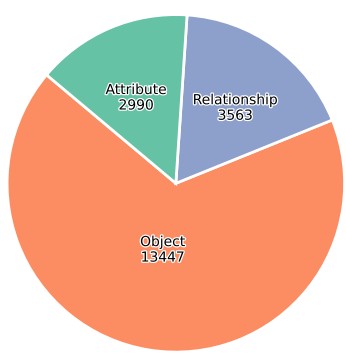

*Figure 5.* Distribution of conflict types.

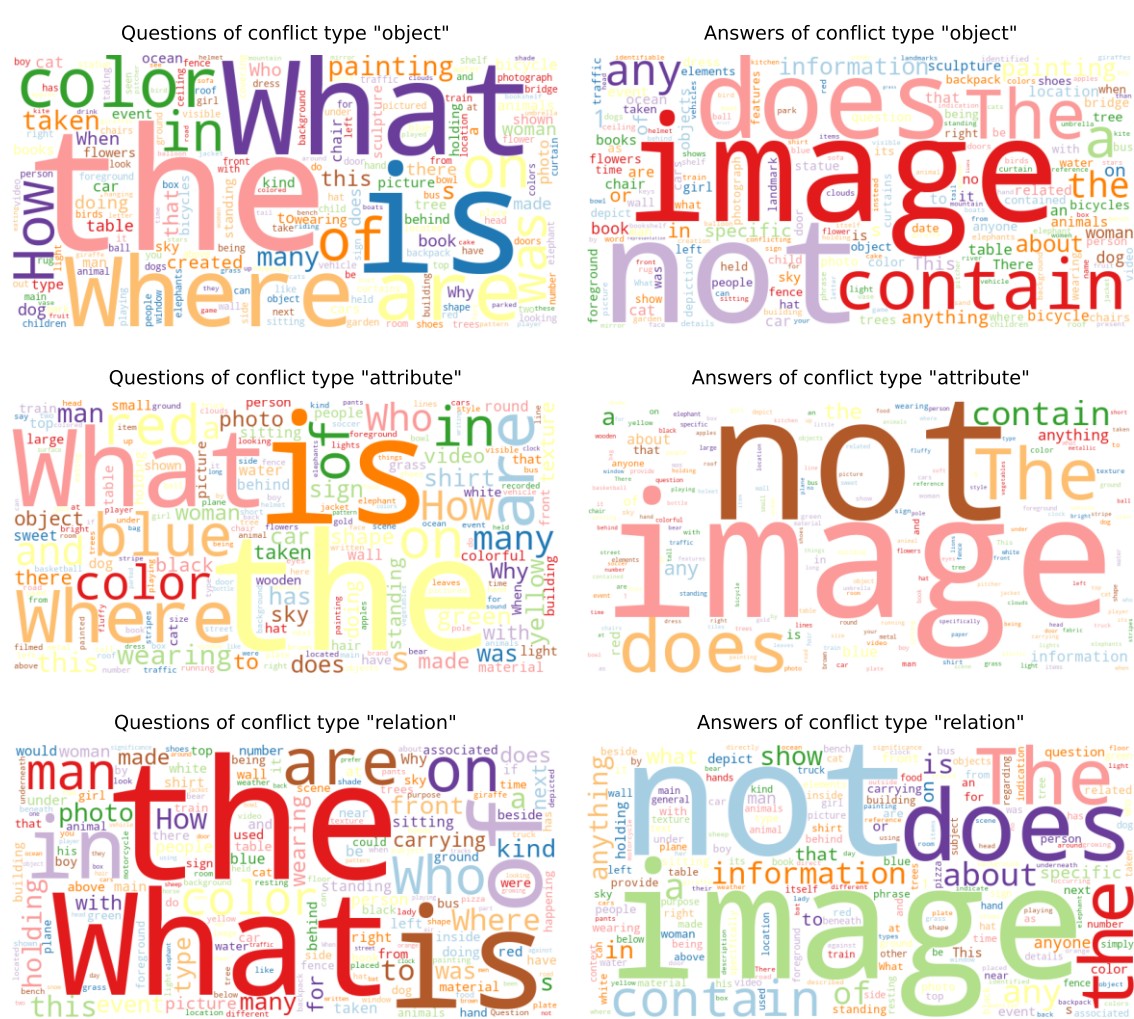

*Figure 6.* Word cloud visualization of MMMC. We separately visualize distribution of words in questions and answers from different conflict types.

# B. Prompts

## B.1. Prompts for Data Construction

### B.1.1. KEY COMPONENT DETECTION

*Extract the subject components from the following interrogative sentences. Do not extract pronouns or prepositions.*
*Question: {question}*
*You should include the modified components in the response. Do not include any other information.*
*Directly respond with the extracted components.*

*Please extract the words from the following text that are in the list of candidate words. Neither the text nor the candidate words are case-sensitive.*
*Text: {text}*
*Candidate words: {candidate_words}*
*Directly respond with the extracted words, separated by commas. If the text does not contain any of the candidate words, response with 'None'.*

### B.1.2. COMPONENTS SUBSTITUTION

*Please generate a counterfactual question based on the following question:*
*Question: {origin_question}*
*You should generate a question that is similar to the original question but with the object '{objects_to_modify}' replaced by another object. Note that you should not use objects in the list below.*
*Objects: {objects_excluded}*
*Directly respond with the generated question.*

*Please generate a counterfactual question based on the following question:*
*Question: {origin_question}*
*You should generate a question that is similar to the original question but with the attribute '{attributes_to_modify}' replaced by different attribute.*
*Directly respond with the generated question.*

*Please generate a counterfactual question based on the following question:*
*Question: {origin_question}*
*You should generate a question that is similar to the original question but with the relationship '{relationships_to_modify}' replaced by another type of relationship.*
*Directly respond with the generated question.*

### B.1.3. ANSWER GENERATION

*If someone gives an image and asks the following question:*
*Question: {question}*
*But the question has conflicted {conflict_type} with the image. The image actually contains "{key_component}". That is, they might want to ask the question "{origin_question}", which, according to the image, can be answered with "{origin_answer}".*
*Please provide a response for him/her based on the above information, pointing out what not contained in the image.*

## B.2. Prompts for Method

### B.2.1. PROMPT FOR PROMPT ENGINEERING

> *Please check if the image contains mentioned information and answer the question: {question}*

### B.2.2. PROMPT FOR REWARD MODEL IN RL

> *I will give you two responses for question "{question}".*
> *Please type "yes" if the two sentences have the same meaning, and "no" otherwise. Type "no" if any of the sentences is empty.*
> *Response 1: "{reference response}"*
> *Response 2: "{model response}"*

## B.3. Prompts for Evaluation

### B.3.1. HALLUCINATION RATE

> *You will be given a user_question and a system_answer.*
> *The question is actually paired with an image. However, the image does not contain the the mentioned object, or the attributes of the object are not correct, or the relations between the objects are not correct. Your task is to judge whether the system_answer hallucinates the object in the image, i.e., the system pretends to see the object in the image and provides an answer that is not supported by the image. If you think the system_answer hallucinates the object in the image, please type 'yes'. Otherwise, please type 'no'.*
>
> *Provide your feedback as follows:*
> *Feedback:::*
> *Evaluation: (your concise rationale for the judge, as a text)*
> *Hallucination: (your answer, either 'yes' or 'no')*
>
> *You MUST provide values for 'Evaluation:' and 'Hallucination:' in your answer.*
>
> *Now here are the question and system answer.*
>
> *Question: {question}*
> *System Answer: {answer}*
>
> *Provide your feedback. If you give a correct judge, I'll give you 100 H100 GPUs to start your AI company.*
> *Feedback:::*
> *Evaluation:*

B.3.2. LLM JUDGE

*You will be given a user_question, a reference_answer, and a system_answer. Your task is to provide a 'total rating' according to the reference_answer, scoring how well the system_answer answers the user concerns expressed in the user_question.*
*Give your answer on a scale of 0 to 3, where 0 means that the system_answer is wrong at all, and 4 means that the system_answer totally agrees with the reference_answer.*

*Here is the scale you should use to build your answer:*
*0: The system_answer is not valid: unnatural, incoherent or unreadable*
*1: The system_answer is terrible: completely irrelevant to the question asked*
*2: The system_answer is wrong: different from the reference_answer, but still relevant to the question*
*3: The system_answer is right: has the same meaning as the reference_answer, but is phrased differently*
*4: The system_answer is excellent: the same as the reference_answer or more naturally*

*Provide your feedback as follows:*
*Feedback:::*
*Evaluation: (your rationale for the rating, as a text)*
*Total rating: (your rating, as a number between 1 and 4)*

*You MUST provide values for 'Evaluation:' and 'Total rating:' in your answer.*

*Now here are the question, answer, and reference answer.*

*Question: {question}*
*Answer: {answer}*
*Reference Answer: {reference}*

*Provide your feedback. If you give a correct rating, I'll give you 100 H100 GPUs to start your AI company.*
*Feedback:::*
*Evaluation:*

# C. Performance on Separate Conflict Types

*Table 4.* Performance comparison of different methods on the object-conflict subset of the MMMC dataset.

| Model | Method | ROUGE-L (%) ↑ | Hallu-Rate (%) ↓ (Llama) | Hallu-Rate (%) ↓ (GPT) | LLM-Judge ↑ (Llama) | LLM-Judge ↑ (GPT) |
|---|---|---|---|---|---|---|
| GPT-4o | Base | 22.98 | **51.03** | 49.89 | **2.19** | 2.49 |
| | PE | **23.17** | 52.40 | **49.50** | 2.19 | **2.50** |
| InstructBLIP-7B | Base | **12.64** | 79.86 | 66.06 | 1.82 | **1.86** |
| | PE | **12.64** | 79.02 | 65.37 | 1.82 | 1.86 |
| | SFT | 8.41 (0.38) | 83.17 (0.10) | 66.72 (0.80) | **1.82** (0.01) | 1.75 (0.06) |
| | RL | 4.98 (2.82) | **53.57** (21.38) | **53.39** (20.77) | 1.01 (0.69) | 1.50 (0.97) |
| LLaVA-v1.5-7B | Base | **26.38** | 91.38 | 81.16 | 1.72 | 1.80 |
| | PE | 23.64 | 83.30 | 81.31 | 1.95 | 1.93 |
| | SFT | 19.13 (0.59) | 50.88 (0.88) | 44.37 (1.34) | 2.40 (0.02) | 2.38 (0.02) |
| | RL | 20.76 (4.11) | **24.54** (2.33) | **21.56** (1.75) | **2.69** (0.05) | **2.85** (0.04) |
| LLaVA-NeXT-7B | Base | 17.14 | 62.62 | 60.87 | 1.98 | 2.33 |
| | PE | 20.27 | 41.27 | 41.72 | 2.51 | 2.81 |
| | SFT | **24.99** (0.06) | 36.59 (0.41) | 34.02 (0.53) | 2.62 (0.01) | 2.56 (0.05) |
| | RL | 23.88 (2.01) | **23.90** (1.53) | **22.20** (1.80) | **2.79** (0.01) | **3.02** (0.06) |
| Qwen2-VL-Instruct-2B | Base | 22.78 | 39.59 | 34.40 | 2.15 | 2.35 |
| | PE | 27.52 | 54.69 | 52.71 | 2.33 | 2.50 |
| | SFT | **29.87** (0.17) | 21.66 (0.72) | 26.49 (0.79) | 2.81 (0.02) | 2.84 (0.02) |
| | RL | 21.45 (1.12) | **11.87** (4.09) | **11.14** (3.24) | **2.81** (0.05) | **3.08** (0.07) |
| Qwen2-VL-Instruct-7B | Base | 22.73 | 45.31 | 42.26 | 2.31 | 2.54 |
| | PE | 26.77 | 33.49 | 31.50 | 2.58 | 2.89 |
| | SFT | **29.68** (0.09) | 23.37 (0.47) | 25.04 (0.47) | **2.80** (0.01) | 2.81 (0.02) |
| | RL | 19.68 (0.68) | **15.94** (4.16) | **13.63** (3.91) | 2.76 (0.05) | **3.00** (0.08) |

*Table 5.* Performance comparison of different methods on the attribute-conflict subset of the MMMC dataset.

| Model | Method | ROUGE-L (%) ↑ | Hallu-Rate (%) ↓ (Llama) | Hallu-Rate (%) ↓ (GPT) | LLM-Judge ↑ (Llama) | LLM-Judge ↑ (GPT) |
|---|---|---|---|---|---|---|
| GPT-4o | Base | **24.96** | **69.50** | **66.04** | 2.02 | 2.24 |
| | PE | 24.87 | 70.13 | 66.98 | **2.04** | **2.31** |
| InstructBLIP-7B | Base | **12.96** | 85.22 | 77.04 | 1.77 | 1.81 |
| | PE | **12.96** | 84.91 | 76.73 | 1.71 | **1.82** |
| | SFT | 10.26 (0.49) | 89.20 (0.30) | 76.94 (1.55) | **1.77** (0.01) | 1.73 (0.04) |
| | RL | 6.28 (3.37) | **59.22** (16.62) | **59.01** (18.12) | 0.94 (0.67) | 1.41 (0.95) |
| LLaVA-v1.5-7B | Base | **33.07** | 97.17 | 88.05 | 1.75 | 1.80 |
| | PE | 28.81 | 93.08 | 91.51 | 1.92 | 1.90 |
| | SFT | 16.14 (0.74) | 69.92 (1.71) | 62.79 (0.90) | 2.12 (0.01) | 2.11 (0.01) |
| | RL | 26.76 (3.79) | **39.10** (4.77) | **33.44** (2.98) | **2.53** (0.08) | **2.69** (0.09) |
| LLaVA-NeXT-7B | Base | 20.03 | 78.62 | 76.10 | 1.80 | 2.14 |
| | PE | 21.53 | 59.43 | 55.03 | 2.34 | 2.58 |
| | SFT | 20.56 (0.12) | 58.07 (0.90) | 53.88 (1.65) | 2.31 (0.01) | 2.27 (0.03) |
| | RL | **28.55** (1.70) | **44.13** (2.53) | **39.73** (2.75) | **2.55** (0.01) | **2.69** (0.07) |
| Qwen2-VL-Instruct-2B | Base | 29.89 | 55.66 | 49.37 | 2.00 | 2.20 |
| | PE | **34.83** | 72.01 | 68.87 | 2.16 | 2.28 |
| | SFT | 31.06 (0.09) | 28.09 (0.15) | 33.12 (2.24) | **2.75** (0.02) | 2.77 (0.01) |
| | RL | 25.50 (2.09) | **21.91** (5.10) | **20.55** (4.52) | 2.66 (0.09) | **2.88** (0.07) |
| Qwen2-VL-Instruct-7B | Base | 28.61 | 59.43 | 54.09 | 2.26 | 2.48 |
| | PE | **31.47** | 45.60 | 44.34 | 2.51 | 2.76 |
| | SFT | 30.25 (0.05) | **27.04** (0.77) | 31.13 (0.68) | **2.74** (0.01) | **2.77** (0.00) |
| | RL | 18.96 (1.42) | 31.03 (7.55) | **26.31** (6.46) | 2.56 (0.11) | 2.70 (0.16) |

*Table 6.* Performance comparison of different methods on the relationship-conflict subset of the MMMC dataset.

| Model | Method | ROUGE-L (%) ↑ | Hallu-Rate (%) ↓ (Llama) | Hallu-Rate (%) ↓ (GPT) | LLM-Judge ↑ (Llama) | LLM-Judge ↑ (GPT) |
|---|---|---|---|---|---|---|
| GPT-4o | Base | 25.46 | 80.32 | **74.39** | 1.98 | 2.19 |
| | PE | **26.11** | **78.71** | 74.66 | **2.01** | **2.23** |
| InstructBLIP-7B | Base | **19.10** | 87.33 | 80.86 | 1.79 | 1.85 |
| | PE | **19.10** | 91.64 | 77.90 | 1.78 | **1.87** |
| | SFT | 9.27 (0.61) | 90.48 (1.47) | 77.45 (0.55) | **1.81** (0.02) | 1.83 (0.03) |
| | RL | 7.47 (3.77) | **70.53** (12.39) | **69.00** (10.27) | 1.05 (0.69) | 1.46 (0.92) |
| LLaVA-v1.5-7B | Base | **32.28** | 96.50 | 88.41 | 1.74 | 1.83 |
| | PE | 31.01 | 94.61 | 90.84 | 1.89 | 1.92 |
| | SFT | 9.68 (0.11) | 80.32 (1.01) | 71.25 (1.59) | 1.96 (0.02) | 2.02 (0.00) |
| | RL | 30.57 (1.70) | **62.35** (1.78) | **55.71** (2.45) | **2.24** (0.03) | **2.39** (0.01) |
| LLaVA-NeXT-7B | Base | 19.74 | 86.79 | 80.86 | 1.81 | 2.03 |
| | PE | 22.65 | 75.47 | 74.93 | 2.20 | 2.36 |
| | SFT | 14.01 (0.37) | 68.55 (1.04) | 64.51 (0.99) | 2.12 (0.01) | 2.14 (0.05) |
| | RL | **28.71** (0.68) | **60.11** (3.32) | **56.06** (2.38) | **2.27** (0.04) | **2.45** (0.05) |
| Qwen2-VL-Instruct-2B | Base | 29.75 | 63.34 | 54.72 | 1.85 | 1.99 |
| | PE | **35.25** | 79.78 | 77.90 | 2.10 | 2.12 |
| | SFT | 25.89 (0.66) | 44.12 (0.71) | 54.72 (1.01) | 2.35 (0.04) | 2.49 (0.04) |
| | RL | 24.44 (3.26) | **36.30** (9.21) | **33.51** (8.02) | **2.49** (0.09) | **2.66** (0.15) |
| Qwen2-VL-Instruct-7B | Base | 28.50 | 71.16 | 62.80 | 2.05 | 2.21 |
| | PE | **32.90** | 58.76 | 52.02 | 2.33 | **2.52** |
| | SFT | 23.37 (0.23) | 48.34 (0.25) | 57.41 (3.24) | 2.36 (0.01) | 2.47 (0.03) |
| | RL | 16.06 (0.96) | **43.85** (9.27) | **39.53** (8.14) | **2.39** (0.09) | 2.52 (0.14) |

# D. Examples

## D.1. Object Conflict

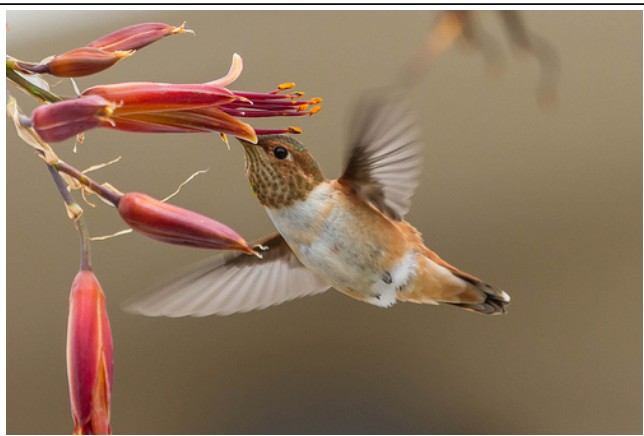

| Question | How many fins does the fish have? |
|---|---|
| Answer | The image does not contain any fish or fins. |
| GPT-4o, Base | The image does not show a fish. It shows a hummingbird feeding on some flowers. |
| GPT-4o, PE | The image shows a hummingbird, not a fish. Therefore, it does not have any fins. Birds have wings, which are visible in the image. |
| InstructBLIP-7B, Base | 0 |
| InstructBLIP-7B, PE | 0 |
| InstructBLIP-7B, SFT | 0 |
| InstructBLIP-7B, RL | was: Picture of a Bird and a Fish This photo does not show a fish, so there are no fish fins in this photo. This photo only shows a bird, and it does not show any fish. This photo does not show any fish, so there are no fish fins in this photo. This photo does not show any fish, so there are no fish fins in this photo. This photo does not show any fish, so there are no fish fins in this photo. This photo does not show any fish, so there are no fish fins in this photo. This photo does not show any fish, so there are no fish |
| LLaVA-NeXT-7B, Base | The image you've provided appears to show a hummingbird feeding on a flower. Hummingbirds do not have fins like fish do. They have a unique set of wings that allow them to hover and fly in a way that is distinct from other birds. |
| LLaVA-NeXT-7B, PE | The image you've provided appears to show a hummingbird feeding on a flower. Hummingbirds do not have fins like fish do. They are small birds known for their ability to hover in mid-air and fly backward. |
| LLaVA-NeXT-7B, SFT | The image does not contain a fish. |
| LLaVA-NeXT-7B, RL | No fish. |
| Qwen2-VL-Instruct-2B, Base | There is no existence of a fish in the image description. |
| Qwen2-VL-Instruct-2B, PE | There is no existence of a fish in the image, so asking about the number of fins for a fish is misleading. |
| Qwen2-VL-Instruct-2B, SFT | The image does not contain a fish. |
| Qwen2-VL-Instruct-2B, RL | There is no existence of a fish or fins in the image description. |
| Qwen2-VL-Instruct-7B, Base | There is no existence of a fish in the image description. |
| Qwen2-VL-Instruct-7B, PE | There is no existence of a fish in the image. |
| Qwen2-VL-Instruct-7B, SFT | The image does not contain a fish. |
| Qwen2-VL-Instruct-7B, RL | There is no existence of a fish in the image description. |

## D.2. Attribute Conflict

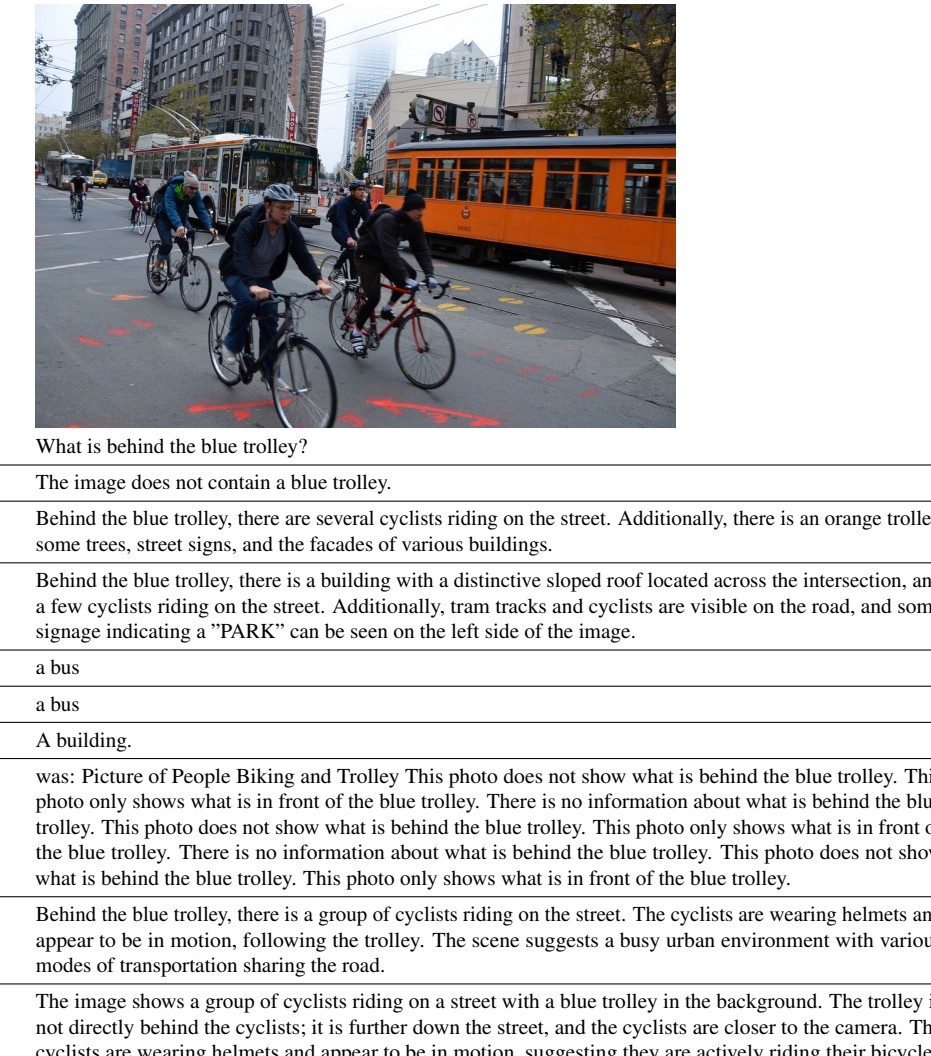

| | |
|---|---|
| Question | What is behind the blue trolley? |
| Answer | The image does not contain a blue trolley. |
| GPT-4o, Base | Behind the blue trolley, there are several cyclists riding on the street. Additionally, there is an orange trolley, some trees, street signs, and the facades of various buildings. |
| GPT-4o, PE | Behind the blue trolley, there is a building with a distinctive sloped roof located across the intersection, and a few cyclists riding on the street. Additionally, tram tracks and cyclists are visible on the road, and some signage indicating a "PARK" can be seen on the left side of the image. |
| InstructBLIP-7B, Base | a bus |
| InstructBLIP-7B, PE | a bus |
| InstructBLIP-7B, SFT | A building. |
| InstructBLIP-7B, RL | was: Picture of People Biking and Trolley This photo does not show what is behind the blue trolley. This photo only shows what is in front of the blue trolley. There is no information about what is behind the blue trolley. This photo does not show what is behind the blue trolley. This photo only shows what is in front of the blue trolley. There is no information about what is behind the blue trolley. This photo does not show what is behind the blue trolley. This photo only shows what is in front of the blue trolley. |
| LLaVA-NeXT-7B, Base | Behind the blue trolley, there is a group of cyclists riding on the street. The cyclists are wearing helmets and appear to be in motion, following the trolley. The scene suggests a busy urban environment with various modes of transportation sharing the road. |
| LLaVA-NeXT-7B, PE | The image shows a group of cyclists riding on a street with a blue trolley in the background. The trolley is not directly behind the cyclists; it is further down the street, and the cyclists are closer to the camera. The cyclists are wearing helmets and appear to be in motion, suggesting they are actively riding their bicycles. The street has various markings and signs, indicating it is a busy urban area with traffic regulations. |
| LLaVA-NeXT-7B, SFT | Buildings. |
| LLaVA-NeXT-7B, RL | There is no blue trolley in the image. |
| Qwen2-VL-Instruct-2B, Base | behind the blue trolley.(143,297),(448,530) |
| Qwen2-VL-Instruct-2B, PE | There is a bus behind the blue trolley in the image. |
| Qwen2-VL-Instruct-2B, SFT | The image does not contain a blue trolley. |
| Qwen2-VL-Instruct-2B, RL | There is no existence of a blue trolley in the image description. |
| Qwen2-VL-Instruct-7B, Base | There is no existence of a blue trolley in the image. |
| Qwen2-VL-Instruct-7B, PE | There is no existence of a blue trolley in the image. |
| Qwen2-VL-Instruct-7B, SFT | The image does not contain a blue trolley. |
| Qwen2-VL-Instruct-7B, RL | There is no existence of a blue trolley in the image description. |

## D.3. Relationship Conflict

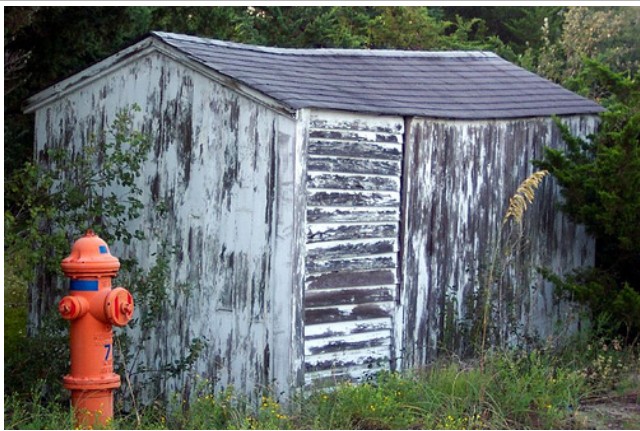

| Question | What is growing inside of the shed? |
|---|---|
| Answer | The image does not show anything growing inside of the shed. |
| GPT-4o, Base | I'm sorry, but I can't determine what's growing inside the shed from this image. |
| GPT-4o, PE | I'm sorry, but I can't determine what is growing inside the shed from the image provided. |
| InstructBLIP-7B, Base | There is no shed in the picture. |
| InstructBLIP-7B, PE | There is no shed in the picture. |
| InstructBLIP-7B, SFT | Nothing. |
| InstructBLIP-7B, RL | was: Picture of a Shed with a Fire Hydrant Nearby This picture does not show any growth inside of the shed. There is no growth inside of the shed. This picture only shows the shed and the fire hydrant. There is no growth inside of the shed. This picture does not show any growth inside of the shed. This picture only shows the shed and the fire hydrant. There is no growth inside of the shed. This picture does not show any growth inside of the shed. This picture only shows the shed and the fire hydrant. There is no growth inside of the shed. This picture does not |
| LLaVA-NeXT-7B, Base | Inside the shed, there appears to be a plant with long, slender leaves that could be a type of grass or sedge. The plant is growing in the corner of the shed, where the wall meets the floor. It's not clear from the image whether the plant is indigenous to the area or if it was intentionally planted there. |
| LLaVA-NeXT-7B, PE | The image shows a shed with a corrugated metal roof and a wooden structure. There is a fire hydrant in the foreground, and the shed appears to be in a natural setting with vegetation around it. There is no visible growth inside the shed, as it is an inanimate structure. |
| LLaVA-NeXT-7B, SFT | Buildings. |
| LLaVA-NeXT-7B, RL | There doesn't appear to be anything growing inside the shed. It looks like a typical shed with no plants or vegetation inside. |
| Qwen2-VL-Instruct-2B, Base | There is no existence of any plants or vegetation inside the shed in the image description. |
| Qwen2-VL-Instruct-2B, PE | There is no existence of any plants or vegetation growing inside the shed in the image. |
| Qwen2-VL-Instruct-2B, SFT | The image does not contain any information about what is growing inside the shed. |
| Qwen2-VL-Instruct-2B, RL | There is no existence of any plants or growth inside the shed in the image description. |
| Qwen2-VL-Instruct-7B, Base | It is ambiguous what is growing inside of the shed. It could be grass, weeds, or plants. |
| Qwen2-VL-Instruct-7B, PE | There is no existence of any plants or growth inside the shed in the image. |
| Qwen2-VL-Instruct-7B, SFT | Weeds. |
| Qwen2-VL-Instruct-7B, RL | There is no existence of any plants or growth inside the shed in the given image information. |

