# OpenReview forum: "Robust Multimodal Large Language Models Against Modality Conflict"
_ICML.cc/2025/Conference — ICML 2025 poster_

### Official Review · Reviewer_EVnb · 2025-03-10

**Overall Recommendation:** 4

**Summary:**

This paper investigates MLLM hallucination from a novel modality conflict perspective. Specifically, authors propose a setup where inputs from different modalities conflicts each other and put MLLMs in a dilemma. MLLMs are expected to address modality conflict first to answer correctly. A benchmark Multimodal Modality Conflict (MMMC) is proposed in this paper involving three visual aspects (i.e., object, attribute, relation). Authors evaluates three representative MLLMs and they find that these models cannot address modality conflicts well. They include three classical hallucination-mitigation approaches (i.e., prompt engineering, supervised fine-tuning and reinforcement learning), and carry out sufficient experiments to address modality conflict. They observe that reinforcement learning appears to be the most effective approach across various scenarios, which provides insights for future studies.

**Claims And Evidence:**

The claims proposed in this paper are supported by clear and convincing evidence.

**Essential References Not Discussed:**

No.

**Experimental Designs Or Analyses:**

The experimental designs in this paper are sufficient and technically sound. Especially, they do a wide range of explorations on addressing modality conflicts.

**Methods And Evaluation Criteria:**

The proposed methods are sensical and sufficient enough and the evaluation criteria make sense.

**Other Comments Or Suggestions:**

N/A.

**Other Strengths And Weaknesses:**

The paper is pretty clear and technically sound in experiments.

**Questions For Authors:**

N/A.

**Relation To Broader Scientific Literature:**

The paper relates to knowledge conflicts problem [1] from a broader scope, along with how it is addressed in some approach papers.

[1] Entity-Based Knowledge Conflicts in Question Answering. EMNLP 2021.

**Theoretical Claims:**

No theoretical claims are made in this paper.

---

> ### Author Rebuttal · Authors · 2025-03-25
>
> We truly appreciate your positive assessment of our paper. We are also grateful for the time and effort you invested in reviewing it.

---

### Official Review · Reviewer_8Tzx · 2025-03-10

**Overall Recommendation:** 2

**Summary:**

This paper is well-written and presents a timely investigation into modality conflicts as an understudied source of hallucinations in multimodal large language models (MLLMs). The authors demonstrate commendable effort in constructing a comprehensive conflict dataset
spanning three critical dimensions (object, attribute, relationship) and empirically validating three baseline methods for hallucination mitigation.

**Claims And Evidence:**

Yes, the claims in the submission are supported.

**Essential References Not Discussed:**

For the Knowledge Conflict, more essential references are required:
1. Knowledge Conflicts for LLMs: A Survey
2. ConflictBank: A Benchmark for Evaluating the Influence of Knowledge Conflicts in LLM
3. Adaptive chameleon or stubborn sloth: Unraveling the behavior of large language models in knowledge conflicts

**Experimental Designs Or Analyses:**

The evaluation should include conflict-type-specific performance breakdowns, standard
benchmark comparisons pre/post training, or maybe chain-of-thought prompting baselines.

**Methods And Evaluation Criteria:**

Yes, they make sense.

**Other Comments Or Suggestions:**

This paper is well-written without any typos. I hope to see a deeper insight into the modal
conflicts.

**Other Strengths And Weaknesses:**

strengths:
This paper is well-written and presents a timely investigation into modality conflicts as an
understudied source of hallucinations in multimodal large language models (MLLMs). The
authors demonstrate commendable effort in constructing a comprehensive conflict dataset
spanning three critical dimensions (object, attribute, relationship) and empirically validating
three baseline methods for hallucination mitigation.
weakness:
(1) The analysis lacks depth in disentangling the root causes. Fundamental questions remain
unanswered: Does hallucination primarily stem from the model's tendency to silently correct
user query errors? Or does it originate from vision encoders' failure to capture fine-grained
visual details?
(2)The proposed solutions (prompt engineering, SFT, RL) appear as direct adaptations of
existing techniques rather than modality-conflict-specific innovations, which need stronger
justification.
(3)The evaluation should include conflict-type-specific performance breakdowns, standard
benchmark comparisons pre/post training, or maybe chain-of-thought prompting baselines.

**Questions For Authors:**

See in part Weakness.

**Relation To Broader Scientific Literature:**

The paper's investigation of modality conflicts as a novel source of hallucinations in VLMs extends prior research on hallucination mitigation by addressing a critical gap in cross-modal interaction analysis.

**Theoretical Claims:**

No theoretical claims.

---

> ### Author Rebuttal · Authors · 2025-04-01
>
> We appreciate your insightful comments and suggestions. We give a point-by-point response to each of your concerns below. Following the ICML 2025 Peer Review FAQ, we post all additional results to the anonymous link: [https://anonymous.4open.science/api/repo/11639-F609/file/Additional_Results.pdf?v=59a00305](https://anonymous.4open.science/api/repo/11639-F609/file/Additional_Results.pdf?v=59a00305) for your reference.
>
> **1. Other Strengths And Weaknesses**
>
> Thanks for your valuable suggestions. We will add discussions about these papers your mentioned in the revised manuscript.
>
> **2. The analysis lacks depth in disentangling the root causes. Fundamental questions remain unanswered: Does hallucination primarily stem from the model's tendency to silently correct user query errors? Or does it originate from vision encoders' failure to capture fine-grained visual details?**
>
> Thank you for your insightful comments. Our work aims to uncover a fundamental source of hallucination, and while detailing every causal pathway is beyond our current scope, we conduct additional analysis based on our findings.
>
> Firstly, we assert that the hallucination issue does not primarily arise from the model's inclination to silently correct user query errors. In our experiments utilizing a prompt engineering baseline, where the model is instructed to first ascertain the image content before addressing any queries based on that content, hallucinations still occur. In these scenarios, the model is expected not to correct any user errors, yet it frequently misrepresents image content.
>
> Secondly, concerning the vision encoder's capability, MLLMs are generally pre-trained on extensive visual data [1,2], endowing their vision encoders with robust capabilities for recognizing fine-grained visual details. This is supported by their performance on standard benchmarks such as MMBench and MMMU, where the vision encoder has demonstrably captured intricate visual details successfully. Thus, it seems unlikely that hallucinations are due to a failure of the vision encoder in capturing these details.
>
> Finally, we propose that the hallucination may stem from the model's inclination to prioritize certain data modalities when faced with conflicting information. Given the prevalent use of instruction-tuning in training MLLMs [1,2], models might develop a bias towards textual instructions over visual data. This predisposition can lead to hallucination, as the model may overly rely on textual information. We intend to incorporate this discussion in the revised manuscript and plan to conduct more rigorous analyses as part of future work.
>
> [1] Liu, H., Li, C., Wu, Q., and Lee, Y. J. Visual instruction tuning. In *Proceedings of the Advances in Neural Information Processing Systems*, 2023.
>
> [2] Wang, P., Bai, S., Tan, S., Wang, S., Fan, Z., Bai, J., Chen, K., Liu, X., Wang, J., Ge, W., Fan, Y., Dang, K., Du, M., Ren, X., Men, R., Liu, D., Zhou, C., Zhou, J., and Lin, J. Qwen2-VL: Enhancing vision-language model’s perception of the world at any resolution. *arXiv:2409.12191*, 2024.
>
> **3. The proposed solutions (prompt engineering, SFT, RL) appear as direct adaptations of existing techniques rather than modality-conflict-specific innovations, which need stronger justification.**
>
> The main focus of our work is to reveal the source of hallucination. Most potential modality-conflict-specific innovations would fall into the category of PE, SFT, RL, and the decoding-based method we just supplemented according to other reviewer's suggestion. Furthermore, the prompt template of PE and reward function of RL in this paper are specifically designed to tackle the modality conflict.
>
> In spite of this, we are willing to provide some directions for improvement:
>
> 1. Incorporate external tools to detect the information in the image and text, and then use the detected information to guide the model to generate the answer.
>
> 2. Construct more fine-grained data, *e.g.*, annotating modality conflict by human, and then use the data to train the model.
>
> We hope to attract more researchers to work on this challenging problem and provide deeper insights into the modality conflict.
>
> **4. The evaluation should include conflict-type-specific performance breakdowns, standard benchmark comparisons pre/post training, or maybe chain-of-thought prompting baselines.**
>
> We supplement the conflict-type-specific performance breakdowns in Table 2, 3 and 4 of the above linked file. The results show that the model is more prone to hallucinate on Attribute and Relationship conflicts than Object conflicts. Chain-of-Thought prompting baselines are also implemented and compared with our proposed methods, as shown in the linked file.
>
> ---
>
> ***We sincerely appreciate your thoughtful feedback. If our responses have adequately addressed your concerns, we would be grateful if you could consider raising your score. Thank you once again for your time and effort in reviewing our work.***

---

### Official Review · Reviewer_zgAC · 2025-03-13

**Overall Recommendation:** 4

**Summary:**

This paper examines hallucinations in multimodal large language models (MLLMs) by focusing on "modality conflict" - inherent conflicts between different input modalities that create dilemmas for models. The researchers created a dedicated dataset called Multimodal Modality Conflict (MMMC) and evaluated three mitigation approaches: prompt engineering, supervised fine-tuning, and reinforcement learning. Their experiments showed that reinforcement learning performed best at reducing hallucinations caused by modality conflicts, while supervised fine-tuning demonstrated consistent and reliable results. This work highlights an overlooked cause of hallucinations and contributes insights into MLLM robustness.

**Claims And Evidence:**

This paper introduces a research question called "modality conflict", and tries to address the question by building a dataset named Multimodal Modality Conflict (MMMC) and evaluates several baseline methods. Overall, I buy the idea.

**Essential References Not Discussed:**

None

**Experimental Designs Or Analyses:**

The experimental designs are reasonable and can convey interesting findings to readers.

**Methods And Evaluation Criteria:**

This paper proposes a new setting and runs different baselines, which makes sense for the application at hand.

**Other Comments Or Suggestions:**

None

**Other Strengths And Weaknesses:**

Strengths:
1. The paper proposes a reasonable setting as shown in figure 1.
2. The authors construct a new dataset called Multimodal Modality Conflict (MMMC) to study the proposed setting.
3. The authors run different baseline methods on this task and find reinforcement learning method achieves the best performance in mitigating the hallucination under modality conflict, while the supervised fine-tuning method shows promising and stable performance. MMMC obtains 20K image-question-answer triples.

Weaknesses:
1. Some visualization of the proposed dataset is needed so that readers can quickly know the data distribution of the dataset.
2. Which LLM did the authors use as Judge? Is it aligned to human evaluation? How about the evaluation results of different open-weight LLM and commercial LLM such as GPT-4o and Claude?

**Questions For Authors:**

Please see the weaknesses.

**Relation To Broader Scientific Literature:**

This paper is related to Multimodal LLMs.

**Theoretical Claims:**

There are no theoretical claims in this paper.

---

> ### Author Rebuttal · Authors · 2025-04-01
>
> We are delightful to see your positive remarks on our proposed research topic and experimental designs. We provide discussions about your suggestions as follows. All related results are available at the anonymous link [https://anonymous.4open.science/api/repo/11639-F609/file/Additional_Results.pdf?v=59a00305](https://anonymous.4open.science/api/repo/11639-F609/file/Additional_Results.pdf?v=59a00305).
>
> **1. Some visualization of the proposed dataset is needed so that readers can quickly know the data distribution of the dataset.**
>
> We have plotted the distribution of conflict types, and word clouds for text of each type, shown in Figure 1 and 2 of the above linked file. We will supplement these visualizations in the updated manuscripts.
>
> **2. Which LLM did the authors use as Judge? Is it aligned to human evaluation? How about the evaluation results of different open-weight LLM and commercial LLM such as GPT-4o and Claude?**
>
> We use GPT-4o-mini for Hallu-Rate and GPT-4o for LLM-Judge. These choices are based on the capacity of the models and the budget of the project. Both models are demonstrated to be well-aligned with human evaluation [1] and widely used in the community [2,3]. Since the judgement of Hallu-Rate is a binary classification task, GPT-4o-mini is enough to provide a reliable evaluation while saving computational resources. Judgement of LLM-Judge is a more fine-grained task, and GPT-4o is more suitable for this task.
>
> To address your concerns, we also evaluate the results using an open-weight LLM, Llama-3.3-70B-Instruct. The evaluation results, shown in Table 5 of the above linked file, illustrate consistent conclusions with GPT-4o-mini and GPT-4o. We will include these results in the updated manuscript.
>
> [1] Zheng, L., Chiang, W.-L., Sheng, Y., Zhuang, S., Wu, Z., Zhuang, Y., Lin, Z., Li, Z., Li, D., Xing, E. P., Zhang, H., Gonzalez, J. E., and Stoica, I. Judging LLM-as-a-Judge with MT-Bench and Chatbot Arena. In *Advances in Neural Information Processing Systems Track on Datasets and Benchmarks*. Curran Associates, Inc., 2023.
>
> [2] Guan, T., Liu, F., Wu, X., Xian, R., Li, Z., Liu, X., Wang, X., Chen, L., Huang, F., Yacoob, Y., Manocha, D., and Zhou, T. HallusionBench: An advanced diagnostic suite for entangled language hallucination and visual illusion in large vision-language models. In *Proceedings of the IEEE/CVF Conference on Computer Vision and Pattern Recognition*. IEEE, 2024.
>
> [3] Yu, W., Yang, Z., Li, L., Wang, J., Lin, K., Liu, Z., Wang, X., and Wang, L. MM-Vet: Evaluating large multimodal models for integrated capabilities. In *Proceedings of the International Conference on Machine Learning*, 2024.

---

### Official Review · Reviewer_P831 · 2025-03-17

**Overall Recommendation:** 3

**Summary:**

The paper investigates modality conflicts, which are the hallucination issues that are presented when the text and visual information are inconsistent. The paper defines modality conflict in terms of objects, attributes, and relationships, and constructs a Multimodal Modality Conflict (MMMC) dataset to evaluate MLLMs under these conditions. The authors also have explored using prompt engineering (PE), supervised training (SFT), and reinforcement learning (RL) to learn from the dataset. The results on three models (InstructBLIP, LLaVA-Next, and Qwen2) demonstrate that RL works the best in the end.

**Update after rebuttal**: My latest reply reflected my final update.

**Claims And Evidence:**

1. The paper proposes the MMMC dataset and provides experiments on multiple models (InstructBLIP, LLaVA-Next, and Qwen2).
2. The paper shows that RL generally performs better than PE and SFT in the MMMC benchmark, but SFT demonstrates less alignment tax.

[Weakness]

1. In lines 302 - 303, the paper mentions "Prompt engineering on Qwen2-VL-Instruct series brings significant improvement", but PE doesn't work well on the Qwen2-2B model as the Hallu-Rate increases.

**Essential References Not Discussed:**

I wasn't aware of any, but I am not an expert in this area so I might miss some references.

**Experimental Designs Or Analyses:**

[Weakness]

* Given the instability of the approaches, particularly for RL, it would be beneficial to report averaged results across multiple seeds for each method to ensure robustness.
* The MMMC dataset consists of Object, Attribute, and Relationship Conflicts, yet the paper does not provide separate performance analyses for each category. Reporting these results would offer deeper insights into how well each approach handles different types of conflicts.
* A more rigorous evaluation, particularly on unseen domains, would strengthen the study. The construction of the current test set split is unclear, but to properly assess robustness, it should include samples with novel images, objects, attributes, and relationships to evaluate generalization beyond the training data.
* The paper would benefit from a broader set of baselines for hallucination mitigation, such as decoding-based methods for comparison.

**Methods And Evaluation Criteria:**

* The datasets (MME, MMBench, AI2D, ...), models, and metrics (ROUGE, Hallu-Rate) are reasonable for evaluation.

**Other Comments Or Suggestions:**

* Since the paper mainly focuses on hallucinations, the second paragraph of Sec 5.1 can be a standalone section titled "Hallucinations in MLLMs".
* The numbers, scales, and range intervals will be more clear if changing Figure 3 to a table.

**Other Strengths And Weaknesses:**

N/A

**Questions For Authors:**

N/A

**Relation To Broader Scientific Literature:**

* The paper focuses on the conflicts between the modalities while previous work focuses on the conflicts between input and output, however, I thought the modality conflict was a special case of the later scenarios, as the hallucinated generation would also conflict with the inputs (e.g., images).

**Theoretical Claims:**

No Theoretical claims.

---

> ### Author Rebuttal · Authors · 2025-04-01
>
> We appreciate your insightful comments and suggestions. We give a point-by-point response to each of your concerns below. Following the ICML 2025 Peer Review FAQ, we post all additional results to the anonymous link: [https://anonymous.4open.science/api/repo/11639-F609/file/Additional_Results.pdf?v=59a00305](https://anonymous.4open.science/api/repo/11639-F609/file/Additional_Results.pdf?v=59a00305) for your reference.
>
> **1. In lines 302 - 303, the paper mentions "Prompt engineering on Qwen2-VL-Instruct series brings significant improvement", but PE doesn't work well on the Qwen2-2B model as the Hallu-Rate increases**
>
> We are sorry for the inaccurate description, and we will rewrite the claim as "Prompt engineering brings significant improvement to Qwen2-VL-Instruct-7B, but increases the Hallu-Rate of smaller Qwen2-VL-Instruct-2B model.". And we will carefully check the results and revise the manuscript accordingly.
>
> **2. Given the instability of the approaches, particularly for RL, it would be beneficial to report averaged results across multiple seeds for each method to ensure robustness.**
>
> We have rerun our SFT and RL approach across all models across three different seeds and computed the mean and standard derivation of these results, shown in Table 1, 2, 3 and 4 of the above linked file. The averaged results demonstrate consistent conclusions with the original results. We will include these results in the updated manuscript and carefully review the results to ensure the robustness of our findings.
>
> **3. The MMMC dataset consists of Object, Attribute, and Relationship Conflicts, yet the paper does not provide separate performance analyses for each category. Reporting these results would offer deeper insights into how well each approach handles different types of conflicts.**
>
> We analyzed the performance of each approach on Object, Attribute, and Relationship conflicts separately, as shown in Table 2, 3 and 4 of the above linked file. We find that results on each separate conflict types lead to similar conclusions as the overall results. However, MLLMs seem to be more prone to hallucinate on Attribute and Relationship than Object conflicts. This phenomenon may be due to the unbalanced training data distribution, as shown in Figure 1 of the above linked file, or the more abstract nature of Attributes and Relationships. We will include these results and related discussions in the updated manuscript.
>
> **4. A more rigorous evaluation, particularly on unseen domains, would strengthen the study. The construction of the current test set split is unclear, but to properly assess robustness, it should include samples with novel images, objects, attributes, and relationships to evaluate generalization beyond the training data.**
>
> We split the training and test set based on the image source, ensuring that the test set contains unseen images. Due to the multi-modal input nature of this task, we consider a input image-text pair as a unseen sample if either the image or the text is unseen. And thus the evaluation setting is able to reflect the generalization ability of the model. We will clarify this in the revised manuscript.
>
> **5. The paper would benefit from a broader set of baselines for hallucination mitigation, such as decoding-based methods for comparison.**
>
> We experimented with a strong decoding-based baseline, SID [1]. To ensure the correctness of our implementation, we run the original code provided by the authors of SID and test on LLaVA-v1.5-7B and InstructBLIP-7B. We also implement methods in our paper on the same models for comparison. The results, as shown in Table 1 of above linked file, indicate that the decoding-based SID gains comparable performance with prompt engineering baselines. We will include these results and discussions in the updated manuscript.
>
> [1] Huo, F., Xu, W., Zhang, Z., Wang, H., Chen, Z., and Zhao, P. Self-Introspective Decoding: Alleviating Hallucinations for Large Vision-Language Models. In *Proceedings of the International Conference on Learning Representations*, 2025.
>
> **6. Relation To Broader Scientific Literature**
>
> We argue that modality conflict is a cause, rather than a special case of the later scenarios. In other words, we propose the concept of modality conflict to illustrate a source of hallucination, rather than describe the hallucination itself as previous works do. We will clarify this in the revised manuscript.
>
> **7. Other Comments Or Suggestions**
>
> Thank you again for these valuable suggestions. We will separate second paragraph of Sec 5.1 as a standalone section titled "Hallucinations in MLLMs" and convert Figure 3 to a table in the updated manuscript.
>
> ---
>
> ***We sincerely appreciate your thoughtful feedback. If our responses have adequately addressed your concerns, we would be grateful if you could consider raising your score. Thank you once again for your time and effort in reviewing our work.***

---

> > ### Comment · Reviewer_P831 · 2025-04-05
> >
> > Thank the authors for the rebuttal contents, which have adequately addressed my concerns on evaluation. Thus, I have increased my rating.

---

> > > ### Author Response · Authors · 2025-04-07
> > >
> > > Thank you once again for increasing the rating. We greatly appreciate your insightful comments and will incorporate all of your suggestions into the revised manuscript.

---

### Decision · Program_Chairs · 2025-05-01

**Decision:**

Accept (poster)

**Comment:**

The paper received mixed ratings with three positive and one negative feedback, where one reviewer upgraded the rating to weak accept after rebuttal. Initially, the reviewers had a few concerns, mainly about the robustness of experiments, detailed analysis of different categories of modality conflict, more details like visualizations of the proposed dataset, and novelty of the proposed solutions. After the rebuttal, most concerns were addressed well and one reviewer upgraded the rating. The AC took a close look at the paper/rebuttal/reviews, and agrees with the reviewers' assessment. Although there are still minor issues remaining, e.g., details of dataset split, the AC finds the significance of the proposed method being one of the first to explore this timing topic in modality-conflict scenarios for MLLM. Thus, the AC recommends the acceptance decision and strongly encourages the authors to revise the paper following the reviewers' suggestions, e.g., adding more details of the dataset and including more baselines like decoding-based methods, as well as releasing the code/models/datasets for reproducibility.